# The Role of Awareness of Consequences in Predicting the Local Tourists' Plastic Waste Reduction Behavioral Intention: The Extension of Planned Behavior Theory

Adel Nasser Badawi [1], Tarek Sayed Adelazim Ahmed [1,2,*], Eid Kaadan Alotaibi [1], Ihab Saad Abbas [1], Ehab Rabee Ali [1] and Eman Sarhan M. Shaker [3]

1    Department of Tourism and Archaeology, College of Arts, University of Hail, Hail 2440, Saudi Arabia; an.badawi@uoh.edu.sa (A.N.B.); e.alotaibi@uoh.edu.sa (E.K.A.); i.saad@uoh.edu.sa (I.S.A.); er.ali@uoh.edu.sa (E.R.A.)
2    Tourist Studied Department, Faculty of Tourism and Hotels, Minia University, El Minya 61111, Egypt
3    Tourist Studies Department, High Institute of Tourism and Hotels-King Mariot, Alexandria 21500, Egypt; habebatarek1982@gmail.com
*    Correspondence: tarekazimh@gmail.com or t.abdelazim@uoh.edu.sa

**Abstract:** Due to increasing concern about plastic waste and its impact on the ecosystem, it is vital to understand tourists' behavioral intentions about plastic waste reduction on beaches. There have been several studies that have used the theory of planned behavior to investigate pro-environmental behaviors or intentions, but there are few specific research studies that have extended the theory of planned behavior by adding awareness of consequences to explain the power of behavioral intention. Accordingly, this paper aimed to investigate how awareness of consequences, subjective norms, attitudes, and behavioral control dimensions influence plastic waste behavioral intention on Jeddah's beaches, in Saudi Arabia. This was performed on a random sample of 390 local tourists in Jeddah city from June to August 2023. A total of 340 of them agreed and answered the questionnaire, yielding a percentage response rate of 87%. This produced 271 valid questionnaires for data analysis after closely examining the survey. A self-complete questionnaire was used for data collection in using multiple statistical analyses to examine the hypotheses. The results demonstrated a positive influence of subjective norms, perceived behavioral control, and consequence awareness on environmental behavioral intention. On the other hand, attitude did not significantly contribute to predict environmental behavioral intention. This study's findings made clear how crucial it is to consider any potential negative effects while making plans to cut down on plastic trash. Educating people about the possible harm that using plastic products on beaches is anticipated to cause to the environment might also be beneficial. It is intended that, through understanding behavior and behavior determinants, governmental bodies, pro-environmental organizations, businesses, and communities will be able to implement appropriate strategies to reduce the use of plastic in Saudi Arabia to protect marine life.

**Keywords:** attitude; subjective norm; awareness of consequences; behavioral intention

## 1. Introduction

Plastic has triumphed in a variety of fields during the previous century, making it pervasive in our daily lives. Plastic products such as plastic bottles and shopping bags have become indispensable in today's lifestyle despite their inventions [1]. However, while plastic is prized for its toughness, its long lifespan poses a problem when it enters the ecosystem and remains as garbage for an extended period of time [2]. Plastic pollution has become a global concern in recent decades, and it is now well known to have negative consequences for both marine species and humans around the world as a result of rising consumerism, urbanization, and changing lifestyles [1]. Plastic waste is one of the most well-known types of beach litter around the world [3]. Plastic pollution is becoming more

of a problem at tourist destinations [4]. Currently, plastic waste is an issue that has grown out of control.

It has an incalculable and permanent negative impact on human health, aesthetics, the economy, public perception, and biological interactions on local, regional, national, and global scales [5]. Plastic pollution in the terrestrial and marine environment is being caused and increased by a lack of legislation, personal and societal behavioral patterns and consumption habits, inefficient use, and bad management [6].

In a world where environmental deterioration is escalating, it is important to understand why people behave in environmentally friendly ways. Because the personal costs of pro-environmental activity are typically far greater than the personal benefits [7], a rational approach to human decision-making predicts that pro-environmental behavior will not be displayed voluntarily [8].

Tourism is one of the most significant sources of waste. Indeed, the effects of tourism on pollution are both obvious and well known [1]. Tourism has been identified as a high-energy, high-water-resource-demanding activity that also produces considerable volumes of solid waste from hotels and recreational areas [9]. The tourism business harms the environment contributing 8% of total greenhouse gas emissions [10] and producing 35 million tons of solid trash yearly [11], including in environmentally sensitive areas. The tourism industry produces 35 million tons of solid trash annually on a global basis [12]. As the tourism industry grows, more garbage is generated from tourism activities and ends up in the ocean because of poor solid waste management after consumption [4].

Tourists can help mitigate this negative impact by planning environmentally friendly vacations and acting in environmentally friendly ways while on vacation [13]. Several studies have been conducted on the impact of tourism on pollution, environmental deterioration, and natural resource depletion. For example, Saenz-de-Miera and Rossello [14] investigated the role of visitors in air pollution in Mallorca, using the number of tourists as an indicator to represent direct and induced environmental pressure [1]. Greiner et al. [15] studied tourist policy and pollution, concluding that in high-pollution conditions, tourism activities may be reduced in the hopes of allowing the ecosystem to recover.

The theory of planned behavior (TPB) is often used to explore the psychological characteristics of pro-environmental behavior [16]. According to the theory, intentions must be formed before actions take place. These intentions can be predicted by three factors: subjective norms, which are expectations or influences from significant others, such as family and friends; attitudes toward the related behavior; and perceived behavioral control, which is the belief that one can control the related behavior [17]. TPB has also been successfully used in the tourism and hospitality industries to predict travelers' pro-environmental behaviors, such as travelers' intentions to stay in eco-friendly hotels when traveling [18,19], travelers' intentions to recycle waste at destinations [20], and travelers' intentions to travel by bicycle [21].

There have been several studies that used TPB to investigate pro-environmental behavior or intention, but there are few specific studies that discuss TPB variables concerning reducing plastic waste behavior on beaches, specifically in Jeddah City. Very little research has been completed on the relationship between behavioral intention and awareness of consequences in the post-consumption context of reducing plastic waste.

Despite the fact that earlier studies have connected travelers to environmentally friendly behavior [22–24], as such, not much is known about how willing tourists are to participate in reducing plastic waste on beaches on vacation. Concurrently, the participation of tourists in reducing plastic waste is expected to enhance the cleanliness and standard of the beach through the completion of tasks. To fill this gap, this study's main objective is to find out how behavioral intention regarding plastic waste in Jeddah, representing Saudi Arabia's beaches, is influenced by awareness of the consequences, subjective norms, attitudes, and behavioral control dimensions.

This study has several contributions as follows:

1. Apart from the previous research contributions, this study stands out as unique being the first attempt in the tourist field to incorporate the theory of planned behavior with awareness of consequences to examine the behavioral intention of local tourists to reduce plastic waste on Jeddah's marine beaches. As a result, awareness is investigated in this work as an antecedent variable that affects behavioral intention. Aside from this condition, it makes perfect sense and reason for the authors to carry out further research on awareness of the consequences of reducing plastic waste on marine beaches.
2. The research's findings complement earlier studies and the related literature on sustainable tourism.
3. The method of establishing people's eco-friendly behavioral intentions to reduce plastic waste on marine beaches was first detailed in Saudi Arabia, which makes this study's findings theoretically valuable.

## 2. Theoretical Background and Hypotheses Development

### 2.1. The Theory of Planned Behavior

The theory of planned behavior is an extension of the theory of reasoned action, necessitated by the original model's inability to deal with behaviors over which humans have only partial volitional control [25,26]. Ajzen [27] added a third factor to the theory of reasoned action (TRA) to account for such restrictions, in addition to attitude and subjective norms. The theory of planned behavior model includes background elements (individual, societal, and information factors) as predictors of behavior in the reasoned action model, which is the most recent version of the reasoned action approach [28].

The TPB model has an advantage over the TRA model in that it can be used to investigate behaviors that are not under voluntary control [29]. The theory of planned behavior (TPB) [30] is a socio-psychological theory that focuses on human behavior research. It has been used to forecast a wide range of tourist behavior. It is a rational decision-making model that predicts behavioral intentions using three important independent factors [31]. The theory of planned behavior postulates three conceptually independent determinants that determine whether a person plans to do something, according to the theory of Ajzen [25]. TPB claims that attitudes, subjective standards, and perceived behavioral control all influence behavioral intentions [32,33]. This theory is used to predict a person's behavioral intentions and behaviors [34].

This social psychology framework proposes that volitional processes, which include the attitudinal dimension (i.e., outcome belief—attitude toward a specific behavior), the normative dimension (i.e., normative belief—subjective norm), and the cognitive dimension (i.e., cognitive belief—cognitive bias), are all interconnected [32]. Rather than focusing solely on the volitional dimension, this socio-psychological theory took into account not only the volitional but also the non-volitional aspects of human decision-making and rational behavior (the theory of planned behavior) [25,33]. In other words, according to the theory of planned behavior, volitional variables cannot adequately account for one's complicated decisions and acts because such decisions and actions are not always under an individual's volitional control [35].

The non-volitional dimension (control belief—perceived behavioral control) influences a person's intention and conduct [32,36]. The theory of planned behavior's core premise is that one's behavioral intention is the most direct and proximate driver of one's conduct [25]. This behavioral intention is generated by the volitional dimension, which includes attitudes toward the behavior and subjective norm, and the non-volitional dimension, which includes perceived behavioral control, according to the theory of Teng et al. [18]. The tourism context provides empirical support for the theory's links between volitional and non-volitional factors and intention, as Lam and Hsu [37] successfully verified that travelers' attitude, subjective norms, and perceived behavioral control significantly influenced their behavioral intentions.

The TPB has a long history of studying the psychological factors that influence pro-environmental behavior [32]. If a person's desire to engage in pro-environmental behavior (PEB) grows sufficiently, they will be able to make more environmentally friendly decisions [38]. In a variety of tourism scenarios, this theory is useful in explaining individuals' ecologically beneficial intentions and behaviors [39].

The direct application of the idea of planned behavior's sufficiency and effectiveness has been questioned numerous times. Environmental awareness, a green image, and anticipated sensations are significant drivers of individuals' different pro-environmental intentions and behaviors, according, in particular, to existing studies in environmental behavior and consumer behavior [40].

De Cannière et al. [33] and Carrus et al. [36] have applied the notion of planned behavior to a wide range of human behaviors. However, because the theory primarily relies on self-identity and self-interest processes, its efficacy in predicting intention and behavior has been questioned [41]. It also ignores how one's intention or decision is energized [42], which is especially important in the context of sustainable tourism [18]. Indeed, several researchers suggested that this approach overlooked some cognitive (e.g., image and awareness) and affective components that are crucial in explaining sustainable/pro-environmental intentions and behaviors [39].

Environmental attitudes, perceived behavioral control, and subjective norms have also been linked to the environmentally respectful behavior when on tourism sites [29]. ERB is also linked to other key aspects of sustainable tourism, including environmental commitment, perceived value, and service quality [43].

People may intend to recycle their household waste, but they do not because they believe that one person's actions will not have a significant environmental impact [44]. A person's desire to engage in pro-environmental behavior (PEB) should grow to the point where they exhibit more favorable behavior toward PEB [45].

### 2.2. Hypotheses Development

### 2.2.1. Attitude and Environmentally Responsible Behavioral Intention

The most applicable definition of attitude was put forward by [25], which is defined as "the degree to which a person has a favorable or unfavorable evaluation or appraisal of the behavior in question". An attitude might be favorable or unfavorable, positive or negative, like or dislike [46]. Since attitude refers to how one feels about performing in a certain way, it can be either positive or negative [47]. A person's desire to engage in or carry out a given conduct is higher the more positive their attitude [48]. It is a complex and multifaceted idea that incorporates both positive and negative environmental perceptions as well as a psychological state that affects individuals' climate-related decisions [49]. Environmental attitudes have a favorable impact on waste classification behavior, according to [50]. The degree to which a consumer has favorable (likes) or unfavorable (dislikes) prospect toward waste reduction actions is referred to as attitude [51]. According to the TPB, attitudes toward particular behaviors positively influence the intention to engage in those behavioral intentions [25,52]. According to the expectation–disconfirmation paradigm, a positive attitude results in favorable expectations, which in turn provide positive motivation to drive behavioral intentions [53]. A person's desire to engage in or carry out a given conduct is higher the more positive their attitude. Environmental attitudes increase a person's propensity to engage in sustainable conduct, according to Hu et al. [54]. Additionally, a person's attitude can predict whether they will act in a pro-environmental manner [55]. Visitors who are more in tune with the local environment are more likely to engage in ecologically responsible behavior [56]. According to Dixit and Badgaiyan [57], people who have a positive attitude are more likely to support sustainable behaviors. Numerous studies have emphasized the correlation between attitude and pro-environmental behavioral intention [19,22,48,58]. According to the empirical research of [58], environmental attitude was positively correlated with intention. Visitors who were more in tune with the local environment were more inclined to practice ecologically responsible behavior [19].

Kumar [59] found that when people had a favorable attitude toward the environment, they would desire to reduce the negative effects of their conduct on the environment. Regarding their research, Hu et al. [54] and Hu et al. [60] claimed that attitudes influence people's intentions to litter. Additionally, Ibrahim et al. [61] investigated the relationship between attitude and anti-littering using data from a survey of 303 Malaysians. The statistical analysis demonstrated that attitude was a strong predictor of anti-littering intention. According to research on waste, attitudes were a significant predictor of travelers' intentions to reduce their waste [46,51,53]. Based on prior research, this study revealed that a more positive outlook increased the intention to act in an environmentally responsible manner [48]. Additionally, Aruta [62] found that favorable views substantially predicted intentions to reduce plastic consumption in order to prevent plastic waste. Higher knowledge had a substantial impact in the discovery of Hajj et al. [63] that favorable attitudes regarding proper (unused or expired) drug disposal were altered. In contrast, Xu et al [64] discovered that attitude had little bearing on one's intention to separate garbage. Based on prior research, this study hypothesized the following:

**H1:** *Local tourists' attitudes toward reducing plastic waste affect positively their behavioral intentions.*

2.2.2. Subjective Norms and Local Tourists' Behavioral Intentions to Reduce Plastic Waste

Subjective norms are the second construct in TPB, which can be defined as "perceived social pressures from referents" such as family members, close friends, and peers [60].

A subjective norm is the perception of important persons who are close to a person and have the power to affect the person's decisions (such as family members, close friends, coworkers, or business partners) [48,65,66]. The norms activation theory [67] proposed that the consumer's subjective norms would imply that performing trash classification was acceptable and valuable because it was compatible with the behavioral patterns of his or her family, friends, and other significant individuals around them [66].

According to the TPB, subjective norms are a major factor in predicting one's intentions [25]. The subjective norm, according to previous researchers, was a crucial variable that affected people's intended pro-environmental conduct [48]. According to Kumar [59], applying more social pressure to individuals may influence their decision to engage in pro-environmental conduct. Friends and family can encourage sustainable lifestyle choices in someone or prevent unsustainable behavior. More social pressure makes people more likely to act in an environmentally responsible manner [68]. Subjective norms, which act as a type of peer pressure, compel people to change their behavior when it comes to being socially and environmentally conscious [69]. Several academics argued that a person's intention to act sustainably could be predicted by the strength of their social pressure [46,48,59,68,70–72].

Numerous empirical studies have shown that subjective norms influence the intention toward ecologically responsible conduct, including the intention to pick up trash [73], prevent littering [54], and separate garbage [64]. According to Venkatesh and Davis [74], a strong predictor of a person's intention to accept a new system is their readiness to live up to the standards of a reference group. Additionally, it has been shown that people will, despite their negative feelings, yield to societal pressure [52]. For instance, it has been discovered that the intentions of outbound visitors to engage in pro-environmental activities are significantly influenced by subjective standards [75]. In the literature on tourism, various studies have demonstrated that tourists were more likely to exhibit environmental behavior if they considered how the reference person wanted them to behave [22,54,60,76]. Numerous research studies in the literature relating to waste have revealed that subjective norms have had an impact on waste minimization intentions [51,53,77]. Additionally, Rakhmawati et al. [78] discovered that social norms significantly influenced visitor waste reduction. However, a study by So et al. [79] found that intentions to reduce plastic trash in Hong Kong were unaffected by subjective standards. Jiang et al. [80] looked at the



psychological factors that influence Chinese farmers' intentions to recycle agricultural waste. They argued that the intention to recycle biomass waste might be significantly stimulated by subjective norms [52]. Khan et al. [68] used the TPB lens to examine behavioral intentions to recycle plastic garbage in a developing context. The findings indicated that subjective norms are significant indicators of consumers' propensity to return.

Fenitra et al. [48] revealed that as societal pressure on tourists increases, so does their intention to act in an environmentally friendly manner. Additionally, a related study that looked at the factors influencing the Zero Litter Initiative (ZLI) at the Huangshan National Park in China supported the findings. Hierarchical regression analysis showed that subjective norms have a favorable impact on intention behavior. Additionally, it was demonstrated that subjective standards had no bearing on individuals' intents to reduce food waste [81]. To counter this claim, Liu et al., Pikturnienė and Bäumle, and Tweneboah-Koduah et al. [22,82,83] stated that raising the subjective norms does not encourage people to act in an environmentally responsible manner. We came up with the second hypothesis in light of the TPB's proposal and the previous discussion.

**H2:** *Local tourists' subjective norms affect positively their behavioral intentions to reduce plastic waste.*

2.2.3. Perceived Behavior Control and Environmentally Responsible Behavior Intention

The perceived behavioral control, which was defined as "the person's perception of the ease or difficulty of performing the behavior of interest" (p.183) [25], is an important factor in determining the TPB [51]. Perceived behavioral control refers to a person's sense of control over their actions and decisions, which determines how they assess the risks and advantages of doing something. As a result, if a person perceives additional difficulties in performing, their desire to do so is reduced [84]. It is concerned with the existence of circumstances that can either facilitate or impede the performance of a behavior [76]. An individual's perceived capacity and ability to carry out a specific behavior is referred to as perceived behavioral control (PBC) [53].

Perceived behavioral control measures an individual's perception of his or her ability to manage volitional elements and foresee difficulties [85]. People's internal controllability and self-efficacy to carry out the behavior, as well as external conditions such as preparation time and facilities, determine the judgment of difficulty [31]. Examples of control factors include the availability or lack of time and money, collaboration with others, the required skills and abilities, and other elements [25]. Mouloudj et al. [51] referred to perceived behavioral control as the person's belief in his or her ability to lower prescription waste. Therefore, impediments to waste reduction behavior can include a lack of facilities for returning undesired prescriptions and the high cost (both financially and psychologically).

This concept was seen by many sustainable behavior researchers to be an important factor in intended behavior [48]. Perceived behavioral control governs intentional behavior by the theory of planned behavior [25]. Numerous studies in the area of tourism have provided empirical support for the positive impact of perceived behavioral control on intentional behavior. According to Lee and Moscardo [86], this concept significantly affected the intention to engage in ecotourism. The research of Han et al. [20] further supported the idea that perceived behavioral control is what motivates tourists to engage in pro-environmental activities. In the context of sustainable behavior, while some studies have demonstrated PBC as having a significant relationship with behavioral intention [72,87], others have discovered minimal or nonexistent correlations [48,59].

The results of these studies have suggested that even if a person has a positive attitude or subjective norms toward the intended act, their behavioral intention will be lower if they have little control over performing a particular behavior due to a lack of available resources (such as financial resources or time) [88]. Perceived behavioral control is the most potent predictor of several pro-environmental behaviors [89]. This is because engaging in pro-environmental behaviors may require some level of personal inconvenience and sacri-

fices [90]. For instance, some researchers discovered that modifying visitors' perceptions of their behavioral control has a beneficial impact on their intention to engage in eco-friendly activities [19]. For example, researchers have discovered that perceived behavioral control was positively correlated with attitudes toward pro-environmental behavioral intentions in a comparative study of visitors' pro-environmental behavioral intentions in destinations with a focus on nature [90]. The results of earlier studies showed a favorable association between PBC and the desire to reduce waste [51,53,77]. For example, consumers' perceived behavioral control positively influenced their intentions to bring a reusable bag when shopping [91]. Additionally, according to a survey of 546 Chinese visitors, perceived behavioral control increased visitors' intention to minimize trash in a good way [53].

Wang et al. [19] conducted additional research that demonstrated that the PBC construct has a significant influence on Chinese customers' decisions to buy environmentally friendly goods. Although these data indicate a positive relationship between perceived behavioral control and the intention to act in an environmentally responsible manner, Pikturnienė and Bäumle and Tweneboah-Koduah et al. [82,83] have found different results. Perceived behavioral control was not linked to a desire to engage in pro-environmental conduct, according to research by [68]. PBC was not a significant predictor of waste separation intention, according to [51,64]. Based on the above, the third hypothesis is as follows:

**H3:** *Local tourists' perceived behavioral control affects positively their behavioral intentions to reduce plastic waste.*

2.2.4. Awareness of Consequences and Environmentally Responsible Behavioral Intention

According to Schwartz [92], environmental awareness is the degree to which a person is mindful of the detrimental effects that their actions may have on their valued objects—such as the environment, other people, animals, habitats, and plants—when they choose not to engage in pro-environmental activities. Awareness of consequences can be defined as "the extent to which someone is aware of the adverse consequences of not acting pro-socially for others or for other things over values" [93]. According to Harland et al. [8], awareness of consequences relates to a person's openness to situational cues of need. Knowing the positive or negative effects of conduct is referred to as awareness of consequences (AC) [94].

Numerous studies in the literature on environmental behavior have highlighted the significant relationships between attitudes toward behavior, environmental awareness, and eco-friendly intentions and actions [35,39,95]. All of these researchers came to the same conclusion: behavioral attitudes are directly impacted by environmental awareness, and behavioral attitudes in turn drive pro-environmental intentions [96]. At six hostels in Byron Bay, Firth and Hing [97] conducted surveys on the attitudes and practices of backpacker hostel visitors in relation to sustainable tourism. It was discovered that the respondents' holiday behavior was allegedly impacted by environmental measures in certain instances. For instance, 17% of participants reported that they had been adopting eco-friendly behaviors, including recycling, while on vacation in the Shire because of Byron Bay's growing environmental consciousness. However, 12% acknowledged that although they were fairly ecologically careful when they were at home, they let this level of care go when they were on vacation. According to [86], guests' positive environmental attitudes may be reinforced by learning about in-resort environmental practices and having positive experiences in ecotourism accommodations, which will pique their interest in more ecotourism activities. According to [98], pro-environmental travel UGC participation was predicated on environmental understanding. Han [24] indicated that awareness of consequences and the normative process were significant predictors of pro-environmental intention. In their examination of young customers' intention formation with regard to green products, Yadav and Pathak [35] identified that environmental awareness was a significant determinant. The results of the study of [39] in the sustainable tourism context also showed the considerable influence of volitional and non-volitional variables on travelers' sustainable intentions and behaviors.

Esfandiar et al. [99] revealed the association between awareness of consequences and personal norms was the strongest and personal norms were the most influential determinants of pro-environmental binning behavior. Hu et al. [60] looked at what influenced visitors' desire to take part in Huangshan National Park's Zero Litter Initiative. The findings showed that understanding visitors' environmental responsibilities is crucial in reducing litter. This theory suggested that there was a strong correlation between visitors' knowledge or awareness levels and littering. Numerous studies have shown that a person's behavior related to littering may be influenced by their level of environmental "awareness" [100,101]. According to research conducted by Heesup Han et al. [96], the theory of planned behavior's predictive value was enhanced by the addition of green images, environmental awareness, and expected sensations. The substantial contribution of these integrated variables to raising intentions for waste reduction was further validated by the results. However, tourism research has highlighted the importance of the influence of characteristics including affective moods, green image, and environmental knowledge on the waste reduction practices of young travelers. Furthermore, rarely does an existing socio-psychological theory for the understandable explanation of young travelers' waste reduction practices when traveling incorporate these crucial ideas [96]. Based on prior research, this study proposes the following hypothesis:

**H4:** *Local tourists' awareness of negative consequences affects positively their behavioral intentions to reduce plastic waste.*

## 3. Research Methodology

The demographics and sample of the research are covered in this section of the paper, along with the instruments and procedures used to gather the data and the statistical methods applied to analyze it. Data were gathered utilizing a self-complete questionnaire, and several statistical analyses were performed to examine the hypotheses. Because of the nature of the data, a quantitative method was used.

### 3.1. Area of Study

Located in the center of Saudi Arabia's eastern coast, Jeddah is widely regarded as the nation's economic and tourism hub [102]. According to Murad [103], the entire area of the municipality is 5460 km$^2$, with an urban boundary of 1765 km$^2$. It is located on the Red Sea shoreline of Makkah Province on the west coast of the nation.

The city of Jeddah is confined to the east by several highland chains that reach an elevation of about 200 m. It is located lengthwise on the shoreline plain and is roughly 10 km wide [104,105]. It is the biggest seaport on the Red Sea and a significant urban hub in western Saudi Arabia. Mecca and Medina are close by, at a distance of only 65 km away [106].

Three millennia ago, during the reign of Othman Bin Affan, the third Muslim Caliph, the city of Jeddah came into being. The latter desired that Jeddah City should develop into a port for receiving Muslim pilgrims, or Hajjis, who traveled from all over the world to Makkah for the Holy Pilgrimage. Since that time, Jeddah has gained recognition as a holy city and as the entrance to the Hijaz, as it serves as the primary air, sea, and pilgrimage route [107]. Jeddah is an important Saudi Arabian city. At just under 4 million as of 2017, it has the second-largest population in the nation [108]. It is among the cities in the nation with the quickest rates of growth [109]. The 110-km-long Jeddah waterfront, also referred to as the Jeddah corniche, is the city's coastal resort area. It is situated beside the Red Sea and stretches approximately 100 km along the Red Sea coast, north to south [110].

The Jeddah corniche is the first outdoor destination for residents and visitors to the city, offering the majority of recreational facilities for sports, enjoyment, and tourism [108]. The corniche is home to the world's tallest fountain, King Fahad's Fountain, as well as a coastal road, pavilions, large-scale public sculptures, and a covered asphalted paved area spanning approximately 3.5 million meters (see Figure 1) [111].

Playgrounds, fun cities, restaurants, buffets, hotels, motels, and beach cabinets are all located along the corniche, ready to welcome guests from Jeddah [112]. The ancient seaport and coastal promenade of Jeddah have long been essential components of the city's urban fabric and cultural identity [113]. It is regarded as the nation's economic and tourism hub [102].

In Saudi Arabia, Jeddah is regarded as a popular tourist destination and a major resort city. Due to its high tourist traffic, the tourism sector is thriving [104,105]. The Globalization and World Cities Study Group and Network [114] designated it as a Beta World City. Due to its religious and historical events, Jeddah is gaining in popularity as a major tourist destination. Additionally, the Kingdom of Saudi Arabia uses Jeddah as a major business hub because it has a port for non-oil-related goods [115].

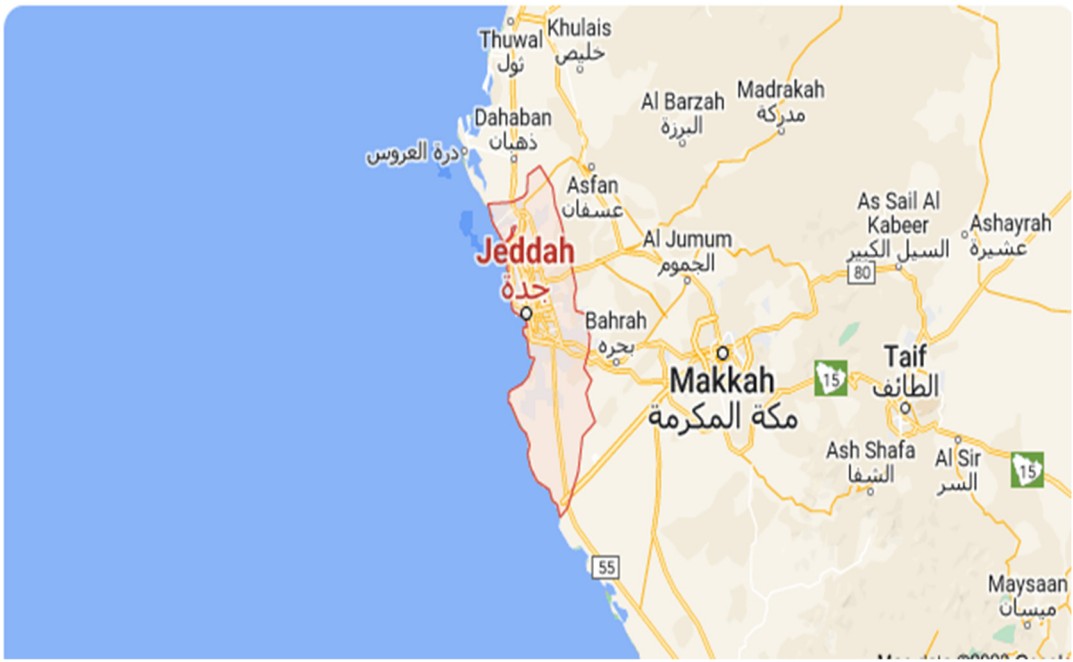

**Figure 1.** Map of Jeddah city [115].

### 3.2. Sample Size

The research population consisted of all local tourists visiting Jeddah beaches. Using the formular of [116], the sample size was determined to be 274 with a 95% confidence level and a 5% margin of error.

Although sample sizes of 200 yield stable results for various fit indices used to measure the degree of fit between the data pattern and the proposed model, Thompson [116] pointed out that there is no standard procedure for determining sample sizes for testing structural equation models. A random technique was used to obtain the sample of local tourists visiting Jeddah city beaches. Using a questionnaire survey, this study used a quantitative method. Based on this, questionnaire surveys were distributed among 390 local tourists who were interviewed over the two-month data collection period between June and August 2023 and requested to participate. The present study's participants were selected randomly. A total of 340 of them agreed and answered the questionnaire, yielding a percentage response rate of 87.17%. This produced 271 valid questionnaires for data analysis after closely examining the survey. The authors requested prior approval from Hail University's Deanship of Scientific Research. Once approved, the primary data collection instrument was a structured questionnaire. A covering letter explaining the purpose, parameters, and confidentiality of this study was attached to every questionnaire. Participation in the survey was strictly voluntary.

### 3.3. Research Instruments

In this study, the questionnaire was divided into two sections. The purpose of the first section of the questionnaire was to gather basic socio-demographic information from the respondents, including gender, age, occupation, education level, and number of beach visits. The purpose of the second section was to measure the research constructs. Every item for each of the five components was modified based on results from earlier studies on environmental behavior. There were a total of 21 items in the measurement set. The attitude items were adapted from [22,25,59]. It was measured using five items, including: "For me, reducing plastic waste in beaches is a good idea (ATT1)"; "I think doing a good job in plastic waste reduction is useful for protecting marine life (ATT2)"; "dropping waste in the beaches has harmful effects on human health (ATT3)"; "I think efforts by official organizations to reduce plastic waste reduction in beaches are effective (ATT4)"; and "I think efforts by official organizations to reduce plastic waste reduction in beaches are important for tourist activity (ATT5)". The subjective norm items were adjusted from [22,25,59] to gauge the social pressure on local tourists to act or not to act on reducing plastic waste. Four items were used to measure this: "I believe that my friends, family, and coworkers expect me to use less plastic when I visit beaches"; "I believe (family, friends, and coworkers) are aware of the pollution caused by plastic waste and tend to reduce plastic waste in their vacations"; "Individuals (friends, family, and coworkers) who are important to me would influence me to reduce plastic waste (SN3)"; and "most people who are important to me would want me to have environmentally responsible behavior (SN4)". The planned behavioral control items were adapted from [22,25,56] to appraise tourists' ability to have self-control while engaging in ERB, including five items: "I am confident that if I want, I can use less plastic materials while I visit beaches (BC1)"; "I have enough time to look for alternatives for plastic materials while visiting beaches (BC2)"; "I have enough money to buy alternatives for plastic materials while I visit the beaches (BC3)"; "It is completely up to me whether or not I can engage in reducing plastic waste in beaches (BC4)"; and "I am confident that if I want, I can have environmentally responsible behavior (BC5)".

Awareness of consequences were adapted from [24,99] including: "The plastic wastes on beaches and their impacts on the marine life are more serious than what individuals think (AC1)"; "I concern that plastic wastes in beaches and their impact on the environment lasts longer than we expect (AC2)"; "I am aware of the seriousness of plastic wastes and their considerable influence on the tourism industry (AC3)"; " If plastic wastes progress due to mass use of plastic in beaches, marine species will become extinct (AC4)"; and "If plastic wastes progress due to mass use of plastic in beaches, environmental threats to public health will become serious (AC5)". The behavioral intention items were adapted from [8,99], including: "I will spend my effort reducing plastic waste when I am on the beach when available in the future (BI1)"; "I have already intended to perform reducing plastic waste behavior (BI2)"; "I am willing to abide by the beaches cleanliness guidelines (BI3)"; and "I will plan to reduce plastic waste behavior rather than disposing of it at will (BI4)".

Every measurement construct was evaluated using a Likert scale with five points. The questionnaire, originally developed in Arabic, was translated into English and then translated back to ensure content validity. In addition, three academics reviewed and commented on it. It was slightly adjusted in terms of wording and formatting.

### 3.4. Data Collection

The data for this study were gathered with a random sampling method. Local tourists visiting Jeddah were asked to engage in the survey on a voluntary basis while on holiday. The technique of data gathering was face-to-face. A pilot study was conducted prior to the final survey to assess the reliability of the questionnaire items. A total of 50 respondents were chosen at random from the target group for this study. Over the two-month data collection period from 15 June to 15 August 2023, data were collected by delivering self-

administered surveys. In total, 390 questionnaires were distributed to the target population of this study.

## 4. Data Analysis

To analyze the constructed model, the current study used the partial least squares structural equation modeling (PLS-SEM) technique. The PLS-SEM was utilized to evaluate the measurement and structural models in this investigation. The measurement model (outer model) relates to the relationship between the constructs and their indicators, whereas the structural model refers to the relationship between the latent constructs themselves. The use of PLS-SEM in this work is due to the fact that it allows for simultaneous analysis of both measurement and structural model, resulting in more accurate calculations.

### 4.1. Characteristics of Respondents

The characteristics of the sample are listed in Table 1. Males made up around 78.2 percent of all respondents. The age range of 21 to 30 represents 55.7% of the respondents. Approximately 75 percent were unmarried. A total of 46.9% of the population had a high school diploma. Of these, 51.8 percent worked. Moreover, 66.4% of participants stated that they visited beaches at least once a week.

**Table 1.** Respondents' demographic characteristics.

| Age Group | Frequency | Percent |
|---|---|---|
| Younger than 20 | 58 | 21.400 |
| 21–30 | 151 | 55.700 |
| 31–40 | 28 | 10.300 |
| 41–50 | 17 | 6.300 |
| 51–60 | 9 | 3.300 |
| 60+ | 8 | 3.000 |
| Total | 271 | 100.000 |
| Gender | | |
| Male | 212 | 78.200 |
| Female | 59 | 21.800 |
| Total | 271 | 100.00 |
| Marital status | | |
| Single | 203 | 74.900 |
| Married | 61 | 22.500 |
| Widow/Widower | 2 | 0.700 |
| Divorced | 5 | 1.800 |
| Total | 271 | 100.000 |
| Education | | |
| Primary School | 000 | 000 |
| Prep. School | 3 | 1.100 |
| High School | 127 | 46.900 |
| University | 118 | 43.500 |
| Post graduate | 18 | 6.600 |
| Other | 5 | 1.800 |
| Total | 271 | 100.000 |

**Table 1.** *Cont.*

| Age Group | Frequency | Percent |
|---|---|---|
| No. of visits to beaches | | |
| 1 times | 180 | 66.400 |
| 1–4 times | 46 | 17.000 |
| 5–10 times | 19 | 7.000 |
| 11–15 times | 26 | 9.600 |
| More than 15 times | 000 | 000 |
| Total | 271 | 100.000 |

*4.2. Means and Standard Deviation*

The descriptive statistics of all the items are shown in Table 2, including their mean and standard deviation. The respondent ranked the following factors as follows: awareness of consequences (mean = 4.628); attitude (mean = 4.332); perceived behavioral control (mean = 3.934); and subjective norms (mean = 3.925).

**Table 2.** Descriptive statistics of questionnaire items.

| | Weight\Scale | Mean Range | Mean | Standard Deviation |
|---|---|---|---|---|
| Attitude | 5 | 4.51–5 | 4.332 | 1.086 |
| Subjective norms | 4 | 3.51–4.50 | 3.925 | 1.317 |
| Perceived behavioral control | 3 | 2.51–3.50 | 3.934 | 1.309 |
| Awareness of consequences | 2 | 1.51–2.50 | 4.628 | 0.636 |
| Behavioral intention | 1 | 1.00–1.50 | 4.675 | 0.566 |

*4.3. Measurement Model Test: Reliability and Validity*

The purpose of this study is to investigate the structural model of ethical conduct among local tourists. The model explains how several latent variables, each with multiple indications, relate to one another. The modeling output, which underwent multiple evaluation phases, is the final model that is being discussed. Initially, the model's validity and reliability for the latent variable construction were assessed.

As shown in Table 3, both the minimum values of CR and Cronbach's alpha are higher than the recommended 0.70, suggesting good internal consistency of the items of each construct. Factor loadings for the constructs and the extracted average variance (AVE) were used to quantify convergent validity. If a construct's AVE value is more than 0.50, it can be accepted and recognized as valid. Given the significance of these requirements, any latent constructs can be deemed sufficient or appropriate. The AVE scores and factor loading data show that all of the measurement items have strong convergent validity. The measuring indicator of the latent variable is legitimate and accepted as a measure of the latent construct if the value of the cross-loading on the variable in question is the biggest and most significant among the cross-loading values for other constructs. Table 4 lists each item's cross-loading that measures the latent variable in this paper. Column 1 has an attitude indicator. The ATT2 indicator appears to have a value of 0.777, while the ATT1 indicator appears to have a value of 0.755. These two indicators have the biggest and most significant loading values when compared to the loading values of the other indicators. This suggests that ATT2 and ATT1 are the best indicators for the attitude construct, particularly when compared to other indicators such as ATT3, ATT4, and ATT5. We can evaluate the subjective norm, perceived behavioral control, awareness of consequences, and behavioral intention constructs' item validity by using the same method.

**Table 3.** Results of measurement model test for reliability and validity.

| Construct | Item | Factor Loading | Cronbach's Alpha | rho_A | Composite Reliability | Average Variance Extracted (AVE) |
|---|---|---|---|---|---|---|
| Attitude | ATT1 | 0.755 | 0.725 | 0.744 | 0.819 | 0.537 |
| | ATT2 | 0.777 | | | | |
| | ATT3 | 0.652 | | | | |
| | ATT4 | 0.604 | | | | |
| | ATT5 | 0.648 | | | | |
| Subjective norm | SN1 | 0.688 | 0.720 | 0.722 | 0.826 | 0.543 |
| | SN2 | 0.753 | | | | |
| | SN3 | 0.714 | | | | |
| | SN4 | 0.790 | | | | |
| Perceived behavioral control | BC1 | 0.772 | 0.792 | 0.815 | 0.855 | 0.543 |
| | BC2 | 0.716 | | | | |
| | BC3 | 0.640 | | | | |
| | BC4 | 0.729 | | | | |
| | BC5 | 0.816 | | | | |
| Awareness of consequences | AC1 | 0.815 | 0.868 | 0.868 | 0.904 | 0.654 |
| | AC2 | 0.800 | | | | |
| | AC3 | 0.818 | | | | |
| | AC4 | 0.779 | | | | |
| | AC5 | 0.832 | | | | |
| Behavioral intention | BI1 | 0.828 | 0.855 | 0.855 | 0.902 | 0.698 |
| | BI2 | 0.864 | | | | |
| | BI3 | 0.868 | | | | |
| | BI4 | 0.779 | | | | |

**Table 4.** Fornell–Larcker criterion (FLC) of all the constructs.

| | Attitude | Awareness of Consequences | Behavioral Control | Behavioral Intention | Subjective Norm |
|---|---|---|---|---|---|
| Attitude | 0.691 | | | | |
| Awareness of consequences | 0.498 | 0.709 | | | |
| Behavioral control | 0.574 | 0.511 | 0.737 | | |
| Behavioral intention | 0.483 | 0.627 | 0.570 | 0.736 | |
| Subjective norm | 0.546 | 0.392 | 0.686 | 0.498 | 0.737 |

The subjective norm construct is best measured by items SN4 and SN2. When compared to other items, items BC5 and BC1 had the greatest loadings (0.816 and 0.772, respectively) for the perceived behavioral control construct, while indications AC1 through AC5 have the highest loadings (0.818 to 0.779) for the awareness of consequences construct. The behavioral intention variable has sufficient loading values for the BI1–BI4 items ranging from 0.868 to 0.779 and for the BI1–BI4 items ranging from 0.828 to 0.779. Measures of each latent construct in the measurement model shown in Figure 2 comprise these helpful items.

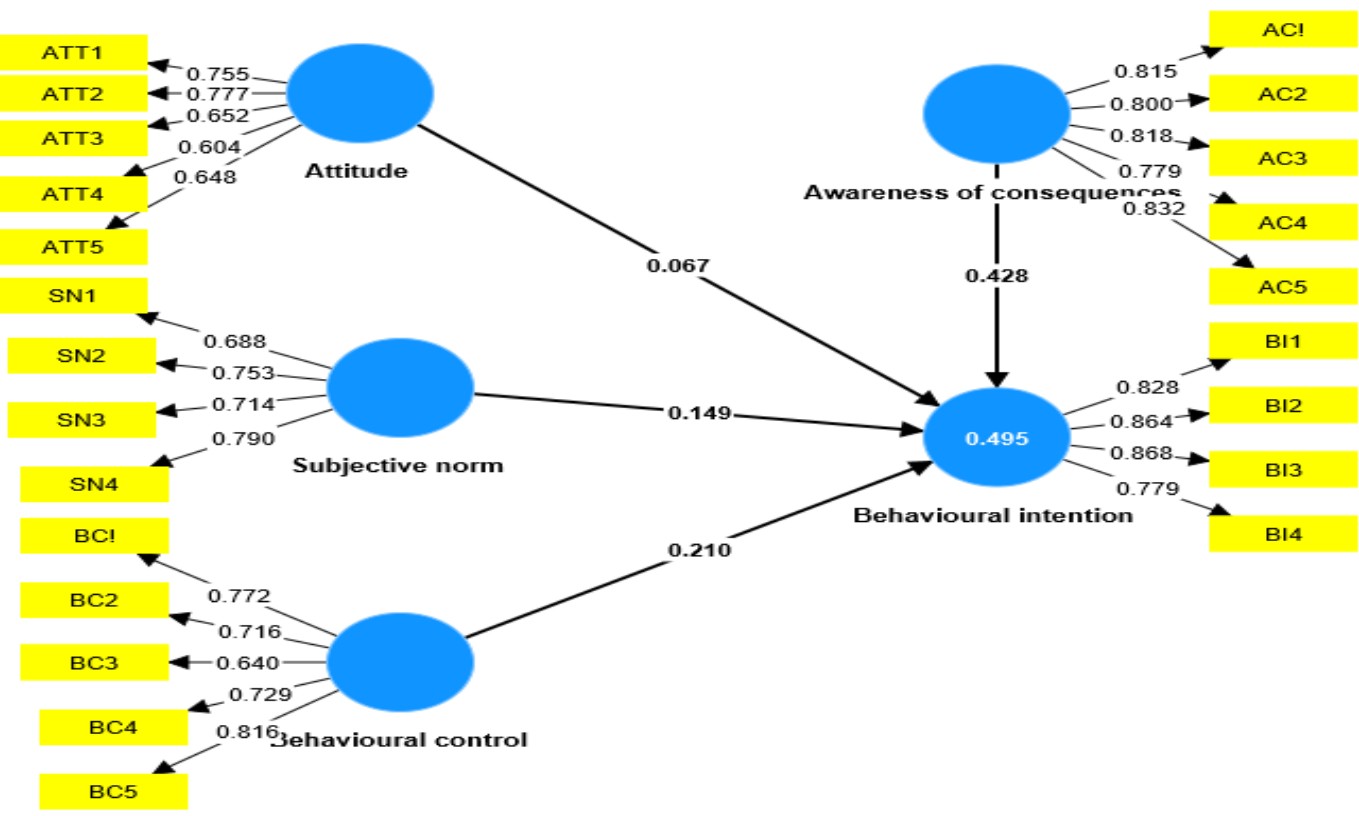

**Figure 2.** Path Analysis Results.

*4.4. Discriminant Validity*

The heterotrait–monotrait ratio (HTMT) and the Fornell–Larcker criteria (FLC) can be used to determine the discriminant validity for each latent construct. Construct multi-collinearity is detected using these two criteria. When a construct's value is less than 0.80, there are no problems with multicollinearity. For each construct, the FLC and HTMT are shown in Tables 4 and 5, respectively. There are no numbers in the off-diagonal cell whose values equal or exceed 0.80. This condition demonstrates that for all latent constructs, multicollinearity is not an issue.

**Table 5.** Heterotrait–monotrait ratio (HTMT) results.

|  | Attitude | Awareness of Consequences | Behavioral Control | Behavioral Intention | Subjective Norm |
|---|---|---|---|---|---|
| Attitude |  |  |  |  |  |
| Awareness of consequences | 0.630 |  |  |  |  |
| Behavioral control | 0.761 | 0.603 |  |  |  |
| Behavioral intention | 0.601 | 0.726 | 0.671 |  |  |
| Subjective norm | 0.774 | 0.488 | 0.912 | 0.625 |  |

*4.5. Path Analysis*

A bootstrapping procedure that examines the statistical significance of the weights of sub-constructs and path coefficients was employed. The hypothesized relationships in the proposed model were evaluated, and the findings from the structural model are shown in Table 6. Hypotheses 1, 3, and 4 propose relationships among the original constructs of TPB theory. Hypothesis 2 proposes an added dimension of awareness of consequences to TPB constructs. Results showed that hypotheses 1, 3, and 4 were supported, and hy-

pothesis 2 was not supported. The findings indicate that attitude value did not predict behavioral intention, while subjective norm, perceived behavioral control, and awareness of consequences values were significant predictors of behavioral intention. Table 6 provides numbers representing the coefficient's amount and the direct relationship between the construct's influence and other constructs. Attitude (t = 0.927 and *p* = 0.354), subjective norms (t = 2.166 and *p* = 0.030), perceived behavioral control (t = 2.325 and *p* = 0.020), and awareness of consequences (t = 5.318 and *p* = 0.000) all had substantial positive effects on pro-environmental behavioral intention except attitude. One's awareness of consequences, behavioral control, and subjective norms increase with improved environmental behavioral intention.

**Table 6.** Hypotheses test results.

| | Original Sample (O) | Sample Mean (M) | Standard Deviation (STDEV) | T Statistics (│O/STDEV│) | *p* Values | Remarks |
|---|---|---|---|---|---|---|
| Attitude -> Behavioral intention | 0.067 | 0.077 | 0.073 | 0.927 | 0.354 | Rejected |
| Awareness of consequences -> Behavioral intention | 0.428 | 0.426 | 0.080 | 5.318 | 0.000 | Accepted |
| Behavioral control -> Behavioral intention | 0.210 | 0.208 | 0.090 | 2.325 | 0.020 | Accepted |
| Subjective norms -> Behavioral intention | 0.149 | 0.150 | 0.069 | 2.166 | 0.030 | Accepted |

## 5. Discussion and Implications

The samples collected in this study were from local tourists' visits to Jeddah city in the Kingdom of Saudi Arabia. The aim of this study is to have a clearer understanding of local tourists' behavior to reduce plastic waste while visiting marine beaches in order to make up for the shortcomings of the existing research. Although previous studies have used the planned behavior theory as a psychological model, this study extended the TPB by adding awareness of consequences to explore pro-environmental behavior regarding plastic waste reduction. Since waste issues are complex and affect several levels of the environment, society, and public health in a country, the data were an excellent fit; the superiority of the proposed model in predicting the behaviors of local tourists to dispose of waste was evident; the predictive relevance of the extended model was empirically identified; and of four assumed relationships, three hypotheses were supported.

Our results revealed that subjective norms exert a significant effect on the intention to reduce plastic waste; it turned out to be an influential construct in forming the intention to reduce plastic waste. This means that reference groups can exert positive pressure on wasteful people to reduce or stop their plastic waste on beaches. As a result, individuals in reference groups who are more trusted have greater persuasive power and are able to put a lot of pressure on others who follow them (i.e., wasters). It is widely accepted and consistent with empirical evidence that subjective norms were an important predictor of intention to reduce waste. This finding is in line with previous works [117,118].

This study demonstrates that planned behavioral control has a positive effect on local tourists' willingness for behavioral intention. This is supported by previous studies [72,87], and it indicates that PBC is accessible through a set of control beliefs that may impede or facilitate behavior. More importantly, it reflects that local tourists are more likely to intend to enact pro-environmental activities to reduce plastic waste. According to some research, within the context of TPB, perceived behavioral control may have the biggest impact on other pro-environmental intentions or behaviors [119].

However, perceived behavioral control has far less of an impact on other pro-environmental actions [35]. The main cause of the various annoyances brought on by various pro-environmental

actions is the varied impact of PBC. Purchasing eco-friendly products, for example, does not require additional work during the entire purchasing process.

On the other hand, findings indicate that attitude does not have a positive effect on plastic waste reduction. This result is in contrast with earlier studies [53,120].

The intention to reduce plastic trash was found to be highly impacted by environmental awareness of the consequences, according to our study. This implies that people will have good intentions to reduce plastic waste when they are more environmentally conscious of the effects and consequences of plastic waste. The majority of respondents obviously know enough about the effects of plastic waste on the environment (M = 4.628). In this regard, Matharu et al. [121] noted that educational initiatives are necessary to increase domestic consumers' awareness of waste. In summary, we find that people who are subject to high-risk perception influences may alter their good intentions about plastic garbage found on Jeddah beaches.

The current study on its own differs from earlier research; as per the authors' expertise, it is among the first studies in tourism research to integrate the theory of planned behavior with awareness of consequences to investigate local tourists' behavioral intention towards reducing waste on the marine beaches of Jeddah. The research's findings complement earlier studies and the related literature on sustainable tourism. The results of this study are theoretically valuable due to the fact that the process of forming individuals' eco-friendly behavioral intentions to reduce plastic waste on marine beaches was first described in Saudi Arabia. Second, this study extended the planned behavior theory by adding the awareness of consequences dimension. Hence, adding awareness of consequences construct to TPB has been proven to be more powerful in predicting local tourists' behavioral intentions.

In other words, local tourists' behavioral intentions are better predicted by AC in conjunction with TPB constructs. Thus, the present study better explains the individuals' eco-friendly behavioral intentions in the context of reducing plastic waste.

This study's findings emphasize the role that awareness of consequences, subjective norms, perceived behavioral control, and perceived behavioral control in persuading visitors to minimize their use of plastic bags on Jeddah's marine beaches. This has a number of managerial implications for destination management. First, this study's findings highlighted how important it is to be mindful of potential drawbacks while developing intentions to reduce plastic waste. It can also be beneficial to inform people about the potential harm that using plastic products on beaches is expected to cause to the environment. It might make local tourists think about the drawbacks of not discarding plastic waste. This study's findings highlighted the role that subjective standards, also known as in-group norms, play in shaping behavioral intentions to reduce plastic waste. It implies that significant others who accompany tourists to their destinations—such as friends, family, and coworkers—should also receive attention.

These significant companions' social pressure can have a big impact on behavioral intention. Therefore, in order to support the promotion and advocacy of plastic waste reduction in tourist locations, awareness campaigns for reductions in plastic trash can be undertaken, during which certain subjective norms might be emphasized. The policymakers and plastic crisis managers are advised to take certain actions, such as giving away free reusable bags to visitors and creating laws that compel visitors and locals to use reusable bags (by, for example, progressively raising the cost of plastic carrier bags). To facilitate the practice of cutting down on plastic waste, perceived behavioral control could be enhanced. According to [122], people are unlikely to use plastic bags if they are not available to them. In order to limit the manufacturing of plastic bags, the government could therefore impose regulations on paid plastic bags or impose a tax on plastic bags. However, the results showed that their behavioral intention toward plastic trash is not predicted by their attitude. Therefore, in order to assist people adopt this new policy concept and establish new habits, the government should step up publicity efforts and provide facilities for the disposal of plastic garbage. People also need to modify the way they live. The government should emphasize more in its programs to reduce plastic garbage because

doing so will help safeguard the environment and positively affect the perceptions of visitors from the area.

## 6. Conclusions

It is true that there is ample evidence of the negative impact of tourism on pollution [1]. It has been determined that tourism uses a lot of energy and water resources. It also generates a significant amount of solid waste from hotels and other tourist destinations [9]. Based on the notion of planned behavior, this study used a structured survey to find out the behavioral intention of 390 local tourists visiting Jeddah beaches to reduce plastic trash. The awareness of consequences aspect was added to the theory of planned conduct in this study. Three main conclusions may be drawn from the results, and they are as follows: The findings indicate that, in terms of path weight to behavioral intention, aware-ness of consequences ranks first, behavioral control ranks second, and subjective norms rank lowest. As a result, when implementing waste reduction, people focus more on their negative issues that arise from discarding plastic waste than on their own abilities and their social networks (friends, family, and universities). Together, these three variables were able to account for almost 77% of the variation in the desire to reduce plastic waste.

Since the awareness of consequences construct accounts for approximately 42% of the variance in plastic waste reduction intention—which is the key determinant of this behavioral intention—it is clear that the TPB model may be extended to account for waste reduction behavior. However, the attitudes of local tourists do not influence their readi-ness to lessen the amount of plastic debris seen on Jeddah beaches. Therefore, given that, the actions of local tourists have the potential to increase the amount of plastic garbage generated, it is worthwhile to look into the elements that lessen this behavior from the per-spective of local tourists in order to comprehend how to involve them in waste management techniques.

According to these findings, educating people about the risks associated with plastic trash, increasing their awareness of the environment, and highlighting the influence of sub-jective norms—like families and relatives—all contribute significantly to the development of intentions to reduce plastic waste. Therefore, it is necessary for policymakers and plastic crisis managers to make this behavior more manageable or convenient for local tourists in order to urge them to limit plastic waste while beaching through possible initiatives including creating reusable bags.

## 7. Limitation and Future Research

In spite of the contributions to this study on tourism sustainability, the following limitations should be carefully taken into account for subsequent investigations: Data for this study were only gathered from one location—Jeddah, Kingdom of Saudi Arabia. It is necessary to repeat these findings in other cities. Consequently, the research findings cannot have a high degree of generalizability. Therefore, more investigation is required to ascertain whether this study's findings may be applied to other areas. Future studies should think about using metropolitan locations in order to increase the research framework's applicability and dependability.

Self-reported data were used to acquire this study's data. To more precisely evaluate the constructs, it is advised that future studies employ multiple data-gathering techniques, numerous metrics, and data triangulation. Second, a questionnaire was used to collect this study's data. The technique was developed by the researchers and consisted of a set of closed-ended questions. There was no provision for respondents to provide answers beyond what was offered. Due to the small sample size, this study did not control for the effects of demographic variables. This makes it possible for more extensive sample sizes to be used in future studies to examine these variables. Lastly, the SEM approach was employed in this study to look at the linear correlations between the variables. The fuzzy-set qualitative comparative analysis can help future researchers better grasp the non-linear effect because it focuses on the asymmetric relationships between variables. This

study was also restricted to assessing how tourists' intentions to engage in ecologically responsible behavior, particularly by cutting back on trash, are formed. Therefore, more research on this process should cover how the purpose behind environmentally responsible conduct is actually converted into such behavior. Future research should also cover other forms of ecologically friendly behavior, such as trash collection, recycling, and nature preservation.

**Author Contributions:** Conceptualization, T.S.A.A., E.S.M.S. and E.R.A.; methodology, I.S.A., T.S.A.A., E.K.A., E.S.M.S. and A.N.B.; software, E.K.A. and T.S.A.A.; validation, T.S.A.A. and E.R.A.; data collection, T.S.A.A., E.S.M.S., A.N.B., E.R.A. and I.S.A.; writing—original draft preparation, T.S.A.A.; writing—review and editing, T.S.A.A., E.R.A., E.S.M.S. and E.K.A.; funding acquisition, A.N.B. All authors have read and agreed to the published version of the manuscript.

**Funding:** This research was funded by the Deanship of Scientific Research at Hail University in the Kingdom of Saudi Arabia [grant number BA-2206].

**Institutional Review Board Statement:** All subjects gave their informed consent for inclusion before they participated in this study.

**Informed Consent Statement:** Informed consent was obtained from all subjects involved in the study.

**Data Availability Statement:** Data available from the corresponding author upon reasonable request.

**Acknowledgments:** The Deanship of Scientific Research at Hail University in the Kingdom of Saudi Arabia provided support for this study. We would therefore like to express our profound gratitude to the whole team of the Deanship of Scientific Research for their outstanding work and unwavering support.

**Conflicts of Interest:** The authors declare no conflicts of interest.

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
