# Peer review of "The Role of Awareness of Consequences in Predicting the Local Tourists’ Plastic Waste Reduction Behavioral Intention: The Extension of Planned Behavior Theory"

_sustainability, doi:10.3390/su16010436_

Round 1
Reviewer 1 Report
Comments and Suggestions for Authors
The paper explores the structural model that explains how awareness of 6 consequences, subjective norms, attitude and behavior control variables influence plastic wastes behavioral intention in Jeddah's beaches, Saudi Arabia. The paper is interesting. The following defects should be improved as follows:
1、 The abstract should be revised to reflect main novelty and contribution of the paper.
2、 In the Introduction, the authors said that “Tourism is one of the most significant sources of waste.”. Please list the data to support your idea.
3、 Why theory of planned behavior (TPB) is appropriate to solve the problem? What’s the main advantage of the method?
4、 In the introduction section, the contribution of the paper should be summarized one by one.
5、 The research gap should be pointed out in the Introduction section.
6、 In section 3.1, Sample size is a little small. Can you extend it?
7、 In section 3.4, To analyze the constructed model, the current study used the Partial Least Squares 460 Structural Equation Modeling (PLS-SEM) technique. Why do you use PLS-SEM to analyze the data?
Comments on the Quality of English LanguageThe Quality of English Language should be improved.
Author Response
|
The abstract should be revised to reflect main novelty and contribution of the paper. |
Authors revised the abstract. We referred to novelty and main contributions. Please see abstract. |
|
In the Introduction, the authors said that “Tourism is one of the most significant sources of waste.” Please list the data to support your idea. |
There is available data about the contribution of tourism as a source of waste. Please see introduction section |
|
Why theory of planned behavior (TPB) is appropriate to solve the problem? What’s the main advantage of the method? |
Please see lines 78-82 and lines 124-125
|
|
In the introduction section, the contribution of the paper should be summarized one by one |
Please see lines 103- 113 |
|
The research gap should be pointed out in the Introduction section. |
Please see lines 94- 100 |
|
In section 3.1, Sample size is a little small. Can you extend it? |
We appreciate the reviewer suggestion but it is not possible to redistribute the questionnaire. |
|
In section 3.4, To analyze the constructed model, the current study used the Partial Least Squares 460 Structural Equation Modeling (PLS-SEM) technique. Why do you use PLS-SEM to analyze the data? |
Please see lines 511-516 |

Reviewer 2 Report
Comments and Suggestions for Authors
The paper is interesting, it deals with an important subject regarding the role of awareness of consequences in predicting the local tourist`s behaviour, in the context of reducing waste and pollution. The paper has merits, is very detailed, well organized, and uses a solid logical tool.
However, in order to improve the quality of the paper, I would suggest several recommendations
The hypotheses, although well framed in the research objectives, sometimes refer to simple statements, close to common sense deductions. Moreover, in some cases the meaning must be clearly specified, avoiding ambiguity E.g, Attitude toward reducing plastic waste affects positively local tourists’ behavioral intentions (whose attitude?)
The literature is well selected, with numerous relevant and up-to-date contributions, correctly and documented supporting the research objectives and issuing working hypotheses. However, its considerable dimensions and poor systematization make it difficult to go through and understand it in the context. We recommend a simple systematization, based on guiding ideas (as sub-sub-chapters).
Formal issues
- Completing the titles of tables 4, 5 (possibly also 6), to be more suggestive and appropriate to the context (not just mentioning the technique)
- At the end of the Results and discussion section (or appropriate) we recommend to insert the centralized situation (tabular or not) of the issued hypotheses – accepted, rejected, partially accepted, etc.
- Capital letter at the beginning of the titles of chapters and subchapters (see 2.1., 2.1.1. 3.3. 3.4 etc), or for the titles of cited journals (in the Final References section)
- Final references have to be carefully reviewed
Ex
- Ajzen, I. (2015). Consumer attitudes and behavior: The theory of planned behavior applied to food consumption decisions. Rivista Di Economia Agrariadi Economia Agraria, 70(2), 121–138. https://doi.org/10.13128/REA-18003 . Journal title is confused (recommended citations Italian Review of Agricultural Economics)
- Fishbein, M., & Ajzen, I. (1977). Belief, attitude, intention, and behavior: An introduction to theory and research. Place and publisher should be added – here (probable) Reading, MA: Addison-Wesley).
Thank you for the opportunity to review this article and good luck!
Author Response
|
The hypotheses, although well framed in the research objectives, sometimes refer to simple statements, close to common sense deductions. Moreover, in some cases the meaning must be clearly specified, avoiding ambiguity E.g, Attitude toward reducing plastic waste affects positively local tourists’ behavioral intentions (whose attitude?) |
Done as follows:
H1: Local tourists ‘attitude toward reducing plastic waste affects positively their behavioral intentions. H2: Local tourists’ subjective norms affect positively their behavioral intentions to re-duce plastic waste. H3: Local tourists perceived behavior control affects positively their behavioral intentions to reduce plastic waste. H4: Local tourists' awareness of negative consequences affects positively their behavioral intentions to reduce plastic waste. |
|
We recommend a simple systematization, based on guiding ideas (as sub-sub-chapters). |
Please see sub sections (done). |
|
Completing the titles of tables 4, 5 (possibly also 6), to be more suggestive and appropriate to the context (not just mentioning the technique) |
We completed titles 4, 5 and 6 as follows Table 4. Fornell–Larcker criterion (FLC) of all the constructs Table 5. Heterotrait–monotrait ratio (HTMT) results Table 6. Hypotheses test results
|
|
At the end of the Results and discussion section (or appropriate) we recommend to insert the centralized situation (tabular or not) of the issued hypotheses – accepted, rejected, partially accepted, etc. |
We added a column to table 6 |
|
Capital letter at the beginning of the titles of chapters and subchapters (see 2.1., 2.1.1. 3.3. 3.4 etc), or for the titles of cited journals (in the Final References section) |
we started the titles of sections and sub sections with capital letters |
|
- Final references have to be carefully reviewed |
All references have been extensively reviewed |
|
Reviewer 3 |
|
|
Please add the information of author's affiliations. |
I think that the editor would add author's affiliations |
|
Line 13, give the full name of TPB when the abbreviation first appears, as does EBI on line 18. |
Done |
|
Line 25, The word "recent" does not feel appropriate for the invention of plastic bags and bottles because they have appeared for a long time in the reader's intuitive sense. |
We deleted the word recent. |
|
Lines 82-85 are unnecessary and suggested to be deleted. |
The last paragraph has been deleted. |
|
Line 88 and Line 159, I am very confused that I did not find any content related to 2.1.2 and 2.2.2 after 2.1.1 and 2.2.1, did the authors miss the relevant part? |
Please see sections and sub sections . |
|
As for the study area, on the one hand, the authors are requested to add the particularity of selecting Jeddah's beaches, Saudi Arabia in the introduction; on the other hand, please supplement the section of “3.1 study area” in Section 3 and provide a map of the study area. |
We added a subsection ( area of study |
|
Line 92, give the full name of TRA when it first appears, the same problem appears in ERB in Line 150. |
Done
|
|
It is suggested to delete some repetitive expressions, such as “In other words, a person's desire to engage in or carry out a given conduct is higher the more positive their attitude” in Line 164-166, and “In other words, a person's desire to engage in or carry out a given conduct is higher the more positive their attitude” in Line 176-177. |
Repetitive expressions have been deleted. |
|
In Section 2.2.1, the authors proposed 4 hypotheses for this study, which is not suitable for the subheading of “2. Literature Review”, it is suggested that this section could be made a separate section. |
We made hypotheses development in a separate sub section. |
|
Table 1, It looks like the last row and the penultimate row are reversed. In addition, please put a header before the marital status information. At the same time, the expression of this table is very confusing, not easy to capture effective information, please modify. |
We corrected all mistakes in table 1 |
|
The numbers after the decimal point in all of the tables in this paper are not uniform, Table 1 has 1 decimal places, table 2 has 5 decimal places, and the rest of the tables have 3 decimal places. Please unify them in the whole manuscript. |
Done |
|
In Table 3, please give the specific information corresponding to ATTI1, ATTI2 and other items, which cannot be obtained in the main text. |
Done. Please see sub section: 3.3. Research instruments.
|
|
Line 536, the authors mentioned Table 7 here, but I did not find any relevant information. |
We are so sorry it is table 6 not 7 |
|
14. Please add the title of Figure 1. |
We added the title of Figure 2. Fig. 2. Path Analysis Results
|
|
Section 6-8 could be merged into the discussion section. |
We merged it into the discussion section |

Reviewer 3 Report
Comments and Suggestions for Authors
The authors investigated how awareness, subjective norms, attitude and behavior control variables influence intention based on a survey. The authors have made some useful explorations, but the method is relatively simple and the interpretation of the results is not deep enough. In addition, there are many errors in the manuscript that should not have occurred, and it seems that the authors did not check before submitting the paper. Therefore, the article needs further modification.
1. Please add the information of author's affiliations.
2. Line 13, give the full name of TPB when the abbreviation first appears, as does EBI on line 18.
3. Line 25, The word "recent" does not feel appropriate for the invention of plastic bags and bottles because they have appeared for a long time in the reader's intuitive sense.
4. Lines 82-85 are unnecessary and suggested to be deleted.
5. Line 88 and Line 159, I am very confused that I did not find any content related to 2.1.2 and 2.2.2 after 2.1.1 and 2.2.1, did the authors miss the relevant part?
6. As for the study area, on the one hand, the authors are requested to add the particularity of selecting Jeddah's beaches, Saudi Arabia in the introduction; on the other hand, please supplement the section of “3.1 study area” in Section 3 and provide a map of the study area.
7. Line 92, give the full name of TRA when it first appears, the same problem appears in ERB in Line 150.
8. It is suggested to delete some repetitive expressions, such as “In other words, a person's desire to engage in or carry out a given conduct is higher the more positive their attitude” in Line 164-166, and “In other words, a person's desire to engage in or carry out a given conduct is higher the more positive their attitude” in Line 176-177.
9. In Section 2.2.1, the authors proposed 4 hypotheses for this study, which is not suitable for the subheading of “2. Literature Review”, it is suggested that this section could be made a separate section.
10. Table 1, It looks like the last row and the penultimate row are reversed. In addition, please put a header before the marital status information. At the same time, the expression of this table is very confusing, not easy to capture effective information, please modify.
11. The numbers after the decimal point in all of the tables in this paper are not uniform, Table 1 has 1 decimal places, table 2 has 5 decimal places, and the rest of the tables have 3 decimal places. Please unify them in the whole manuscript.
12. In Table 3, please give the specific information corresponding to ATTI1, ATTI2 and other items, which cannot be obtained in the main text.
13. Line 536, the authors mentioned Table 7 here, but I did not find any relevant information.
14. Please add the title of Figure 1.
15. Section 6-8 could be merged into the discussion section.
Comments on the Quality of English LanguageMinor editing of English language required.
Author Response
|
Please add the information of author's affiliations. |
I think that the editor would add author's affiliations |
|
Line 13, give the full name of TPB when the abbreviation first appears, as does EBI on line 18. |
Done |
|
Line 25, The word "recent" does not feel appropriate for the invention of plastic bags and bottles because they have appeared for a long time in the reader's intuitive sense. |
We deleted the word recent. |
|
Lines 82-85 are unnecessary and suggested to be deleted. |
The last paragraph has been deleted. |
|
Line 88 and Line 159, I am very confused that I did not find any content related to 2.1.2 and 2.2.2 after 2.1.1 and 2.2.1, did the authors miss the relevant part? |
Please see sections and sub sections . |
|
As for the study area, on the one hand, the authors are requested to add the particularity of selecting Jeddah's beaches, Saudi Arabia in the introduction; on the other hand, please supplement the section of “3.1 study area” in Section 3 and provide a map of the study area. |
We added a subsection ( area of study |
|
Line 92, give the full name of TRA when it first appears, the same problem appears in ERB in Line 150. |
Done
|
|
It is suggested to delete some repetitive expressions, such as “In other words, a person's desire to engage in or carry out a given conduct is higher the more positive their attitude” in Line 164-166, and “In other words, a person's desire to engage in or carry out a given conduct is higher the more positive their attitude” in Line 176-177. |
Repetitive expressions have been deleted. |
|
In Section 2.2.1, the authors proposed 4 hypotheses for this study, which is not suitable for the subheading of “2. Literature Review”, it is suggested that this section could be made a separate section. |
We made hypotheses development in a separate sub section. |
|
Table 1, It looks like the last row and the penultimate row are reversed. In addition, please put a header before the marital status information. At the same time, the expression of this table is very confusing, not easy to capture effective information, please modify. |
We corrected all mistakes in table 1 |
|
The numbers after the decimal point in all of the tables in this paper are not uniform, Table 1 has 1 decimal places, table 2 has 5 decimal places, and the rest of the tables have 3 decimal places. Please unify them in the whole manuscript. |
Done |
|
In Table 3, please give the specific information corresponding to ATTI1, ATTI2 and other items, which cannot be obtained in the main text. |
Done. Please see sub section: 3.3. Research instruments.
|
|
Line 536, the authors mentioned Table 7 here, but I did not find any relevant information. |
We are so sorry it is table 6 not 7 |
|
14. Please add the title of Figure 1. |
We added the title of Figure 2. Fig. 2. Path Analysis Results
|
|
Section 6-8 could be merged into the discussion section. |
We merged it into the discussion section |

Reviewer 4 Report
Comments and Suggestions for Authors
Dear Authors,
First and foremost, I would like to express my appreciation for the effort you have put into your manuscript and for the opportunity to review it. Your research addresses an important topic, and I believe that with some revisions it could make a significant contribution to the field. Below are my suggestions for improvement:
General Comments:
1. Use of past tense: Please ensure that references to previously published material and results of your research are consistently written in the past tense.
2. Abbreviations: All abbreviations should be clearly defined the first time they are used in the text. For example, please clarify abbreviations on line 13 of the TPB and line 18 of the EBI.
3. Sentence length in paragraphs: Some sections, including parts of the abstract (lines 6-19), contain numerous of very short sentences. These could be combined or restructured to improve flow and readability.
4. Citation practices: There are inconsistencies in following citation rules throughout the article. A thorough review and revision is needed.
5. Capitalization and terminology: Please review the manuscript for proper capitalization, especially in headings, and for consistent and accurate use of terminology. For example, the term "random sampling" appears to be misused.
6. Language and Proofreading: The manuscript would benefit from extensive proofreading to address awkward phrasing and improve overall clarity.
Specific comments:
1. Introduction: The introduction seems disproportionately long without adding substantial context to the research questions. A more focused analysis and synthesis of the previous findings cited would be beneficial.
2. Literature Review (lines 88-389): Aim for a concise synthesis of relevant findings rather than a descriptive listing of various studies. This will help maintain the focus and purpose of your communication.
3. On line 393, please clarify what is meant by "descriptive survey design" in the context of your study.
4. Lines 402 and 455 raise concerns about your sampling methodology. Clarify whether it was truly random (based on probability principles) or convenience sampling, and provide details on the sampling frame and procedure.
5. Table 2: Include the range of the scale to better interpret the means, and standardize the formatting of numerical values.
6. Misused terms: On line 521, the term "variable" is used incorrectly. Consider revising it to "construct" or "dimension," as appropriate.
7. Discussion: Begin the discussion section with a summary of the main findings.
In conclusion, I believe that these revisions will greatly improve the clarity and overall quality of your manuscript. Your work is valuable, and I am optimistic about its potential contribution to the field once these improvements are made. I look forward to receiving the revised manuscript.
Sincerely,
Comments on the Quality of English LanguageSome sections, including parts of the abstract (lines 6-19), contain numerous of very short sentences.
Inproper capitalization, especially in headings.
Awkward phrasing.
Author Response
|
Use of past tense: Please ensure that references to previously published material and results of your research are consistently written in the past tense. |
Done. |
|
Abbreviations: All abbreviations should be clearly defined the first time they are used in the text. For example, please clarify abbreviations on line 13 of the TPB and line 18 of the EBI. |
Done |
|
Sentence length in paragraphs: Some sections, including parts of the abstract (lines 6-19), contain numerous of very short sentences. These could be combined or restructured to improve flow and readability. |
Please see the abstract. |
|
Citation practices: There are inconsistencies in following citation rules throughout the article. A thorough review and revision is needed. |
The reviewed extensively all references and we did many changes to citations. |
|
Capitalization and terminology: Please review the manuscript for proper capitalization, especially in headings, and for consistent and accurate use of terminology. For example, the term "random sampling" appears to be misused. |
We started the titles of sections and sub sections with capital letters. |
|
Language and Proofreading: The manuscript would benefit from extensive proofreading to address awkward phrasing and improve overall clarity. |
The manuscript has been proofred. |
|
Introduction: The introduction seems disproportionately long without adding substantial context to the research questions. A more focused analysis and synthesis of the previous findings cited would be beneficial. |
Done. Please see the introduction section |
|
On line 393, please clarify what is meant by "descriptive survey design" in the context of your study. |
Please see the first paragraph in the methodology section. |
|
Lines 402 and 455 raise concerns about your sampling methodology. Clarify whether it was truly random (based on probability principles) or convenience sampling, and provide details on the sampling frame and procedure. |
It was a random sample among local tourists visiting Jeddah city beaches. We used self-admimstred questionnaire
The research population consists of all local tourists visiting Jeddah beaches. Using [124] formula, the sample size was determined to be 274 with a 95% confidence level and a 5% margin of error. Although sample sizes of 200 yield stable results for various fit indi-ces used to measure the degree of fit between the data pattern and the proposed model, [124]. point out that there is no standard procedure for determining sample sizes for test-ing structural equation models. . A random technique was used to obtain the sample of lo-cal tourists visiting Jeddah city beaches. Using a questionnaire survey, this study uses a quantitative method. Based on this, questionnaire surveys were distributed among 390 lo-cal tourists who were interviewed over the two-month data collection period between June and August 2023 and requested to participate. The study's participants were selected ran-domly. 340 of them agreed and answered the questionnaire, yielding a percentage re-sponse rate of 87.17%. This produced 271 valid questionnaires for data analysis after closely examining the survey. |
|
Table 2: Include the range of the scale to better interpret the means, and standardize the formatting of numerical values. |
Done, please see table 2. |
|
Misused terms: On line 521, the term "variable" is used incorrectly. Consider revising it to "construct" or "dimension," as appropriate. |
We replaced the term variable with terms ("construct" or "dimension," |

Round 2
Reviewer 1 Report
Comments and Suggestions for Authors
The paper is revised. But there are still some problems, such as too many references are used. Besides, the main contributions of the paper should be summarized one by one in the introduction section.
Comments on the Quality of English LanguageThe Quality of English Language can be improved.
Author Response
Cover letter
December 15, 2023
Editorial office of Sustainability Journal
Dear Editor of Sustainability Journal,
First, I would like to thank you very much for all your efforts to review our manuscript. I really appreciate all reviewers who spent their precious times to report their comments, which I found very useful to improve the quality of our manuscript. Really, it is one of the best experience.
We are resubmitting our manuscript entitled “The role of awareness of consequences in predicting the local tourist`s plastic waste reduction behavioral intention: The extension of planned behavior theory”.
We did our best to respond precisely to reviewers comments.
Yours Sincerely,
Prof. Tarek Sayed Abdel Azim Ahmed
Hail University
Hail city, Saudi Arabia
Tel.: +966536831817
E-mail: t.abdelazim@uoh.edu.sa
|
Reviewer no.1 |
|
|
Too many references are used. |
First, we really appreciate your good remarks in round 1 and round 2. We added more references to cover the sub section area of study. In addition, we extensively used all related references to compose an enough theoretical background to sustain our conceptual model. |
|
The main contributions of the paper should be summarized one by one in the introduction section. |
We summarized them one by one at the end of the introduction section. Please see lines 96-108. |
|
The Quality of English Language can be improved. |
Our manuscript has been proofread two times. |
|
Reviewer no.2 |
|
|
Several minor formal errors - capital letters at the beginning of the sentence, typo errors - Source instead of Soure, different font size. |
Done : please see the title Please see lines 298-299; 305, 433, 441, 456, 631,677, 679. Please see titles of sections and subsections. |
|
Reviewer no.3 |
|
|
The summary does not need to be divided into several paragraphs. |
- Done. Please see abstract. -
|
|
2. The author's name cannot be referenced directly when appearing at the beginning of a sentence, e.g. Line 65 “For example, [14] investigated the role”, Line 67, “[15] studied tourist policy”, Line 153, “[33, 36] have applied the notion”. There are many similar questions in the article, please check it carefully. |
- All done. We checked all references. Thank you for your good remark. |
|
3. Line 263-266, these two hypotheses appear to be repeated, which one does the author wish to retain? |
We deleted one of these two hypotheses. |
|
4. The section of “3.6 Results” should not be put in the section “3. Research Methodology”. |
Please see 4. Data analysis
|
|
Table 1, the percentage of Widow/ Widower “.7” should be “0.7”. |
Corrected |
|
This is an issue that was raised but not resolved last time, “The numbers after the decimal point in all of the tables in this paper are not uniform, Table 1 has 1 decimal places, table 2 has 5 decimal places, and the rest of the tables have 3 decimal places. Please unify them in the whole manuscript”. |
We unified them. |
|
Figure 1, many numbers and words in the picture are overlapped. |
We replaced the map with another one, which is clearer. |

Reviewer 2 Report
Comments and Suggestions for Authors
The authors carefully and precisely addressed the suggestions and recommendations made in our previous review.
As a result the article is better organized, with increased relevance.
Several minor formal errors - capital letters at the beging of the sentence, typo errors - Source instead of Soure, different font size.
Author Response

(The authors gave the same response as above.)

Reviewer 3 Report
Comments and Suggestions for Authors
After revision, the manuscript still has several problems that worries me. In particular, some of the problems mentioned in last round have not been completely revised, please address the comments bellow carefully.
1. The summary does not need to be divided into several paragraphs.
2. The author's name cannot be referenced directly when appearing at the beginning of a sentence, e.g. Line 65 “For example, [14] investigated the role”, Line 67, “[15] studied tourist policy”, Line 153, “[33, 36] have applied the notion”. There are many similar questions in the article, please check it carefully.
3. Line 263-266, these two hypotheses appear to be repeated, which one does the author wish to retain?
4. The section of “3.6 Results” should not be put in the section “3. Research Methodology”.
5. Table 1, the percentage of Widow/ Widower “.7” should be “0.7”.
6. This is an issue that was raised but not resolved last time, “The numbers after the decimal point in all of the tables in this paper are not uniform, Table 1 has 1 decimal places, table 2 has 5 decimal places, and the rest of the tables have 3 decimal places. Please unify them in the whole manuscript”.
7. Figure 1, many numbers and words in the picture are overlapped.
Author Response

(The authors gave the same response as above.)

Reviewer 4 Report
Comments and Suggestions for Authors
Dear Authors,
All of my previous comments have been adequately addressed. I have no further comments.
Sincerely,
Author Response

(The authors gave the same response as above.)
